# The interhemispheric gradient of $SF_6$ in the upper troposphere

Tanja J. Schuck[1,*], Johannes Degen[1,*], Eric Hintsa[2,3], Peter Hoor[4], Markus Jesswein[1], Timo Keber[1], Daniel Kunkel[4], Fred Moore[2,3], Florian Obersteiner[5], Matt Rigby[6], Thomas Wagenhäuser[1], Luke M. Western[3], Andreas Zahn[6], and Andreas Engel[1]

[1]University of Frankfurt, Institute for Atmospheric and Environmental Sciences, Frankfurt, Germany
[2]Cooperative Institute for Research in Environmental Sciences, University of Colorado Boulder, Boulder, Colorado, United States
[3]NOAA Global Monitoring Laboratory, Boulder, Colorado, United States
[4]Johannes Gutenberg University of Mainz, Institute for Atmospheric Physics, Mainz, Germany
[5]Karlsruhe Institute of Technology, Institute of Meteorology and Climate Research, Karlsruhe, Germany
[6]University of Bristol, School of Chemistry, Bristol, United Kingdom
[*]These authors contributed equally to this work.

**Correspondence:** Tanja J. Schuck (schuck@iau.uni-frankfurt.de)

**Abstract.** Anthropogenic trace gases often exhibit interhemispheric gradients because of larger emissions in the northern hemisphere. Depending on a tracer's emission pattern and sink processes, trace gas observations can thus be used to investigate interhemispheric transport in the atmosphere. Vice versa, understanding interhemispheric transport is important for interpreting spatial tracer distributions and for inferring emissions. We combine several data sets from the upper troposphere (UT) to investigate the interhemispheric gradient of sulfur hexafluoride ($SF_6$) covering latitudes from $\sim 80°$ N to $\sim 60°$ S: canister sampling based measurements from the IAGOS-CARIBIC infrastructure and data from the in-flight gas chromatography instruments GhOST and UCATS. The interhemispheric gradient of $SF_6$ in the UT is found to be weaker than near the surface. Using the concept of a lag time removes the increasing trend from the time series. At the most southern latitudes, a lag time of over 1 year with respect to the northern mid-latitude surface is derived, and lag times decrease over the period 2006–2020 in the extra-tropics and the southern tropics. Observations are compared to results from the two-dimensional AGAGE 12-box model. Based on EDGAR 7 emissions, fair agreement of lag times is obtained for the northern hemisphere, but southern hemispheric air appears too "old". This is consistent with earlier findings that transport from the northern extra-tropics into the tropics is too slow in many models. The influence of the emission scenario and the model transport scheme are evaluated in sensitivity runs. It is found that EDGAR 7 underestimates emissions of $SF_6$ globally and in the southern hemisphere, whereas northern extra-tropical emissions seem overestimated. Faster southward transport from the northern extra-tropics would be needed in the model, but transport from the southern tropics into the southern extra-tropics appears too fast.

# 1 Introduction

Sulfur hexafluoride ($SF_6$) is a long-lived greenhouse gas, which is quantified by a Global Warming Potential over 100 years of 25200 (Smith et al., 2021). It is an anthropogenic trace gas which is primarily used for electrical applications (Hu et al., 2023). The adverse effects on climate are partly related to its long atmospheric lifetime, recently estimated at 850 years (Ray et al., 2017). It does not have any tropospheric sinks but is destroyed by high energy solar radiation and electron attachment in the mesosphere (Kovács et al., 2017). Large-scale industrial usage started in the 1950s, its atmospheric mixing ratios increased steadily over the last four decades and are currently around 11.5 ppt (Ko et al., 1993; Maiss et al., 1996; Levin et al., 2010; Simmonds et al., 2020). Its monotonic increase allows to study transport processes in the stratosphere and troposphere based on the concept of the age of air (AoA) (e. g. Bischof et al., 1985; Volk et al., 1997; Hall and Waugh, 1998; Engel et al., 2009; Patra et al., 2009; Bönisch et al., 2011; Waugh et al., 2013; Ray et al., 2014).

The spatial emission patterns and the long lifetime make $SF_6$ an excellent tracer of atmospheric transport processes. For example, $SF_6$ has also been used to estimate an interhemispheric exchange time (Maiss et al., 1996; Patra et al., 2009; Yang et al., 2019) based on the fact that emissions occur primarily in the northern hemisphere. This leads to a pronounced interhemispheric gradient in the troposphere at all altitudes (Maiss et al., 1996; Gloor et al., 2007; Orbe et al., 2021). Model results indicate that interhemispheric transport is asymmetric, with transport from the northern into the southern hemisphere being faster than vice versa, which influences the north-south gradient and interhemispheric transport times (Krol et al., 2018). Typical values for the tropospheric AoA or mean transit times from the northern into the southern hemisphere are around 1.1 to 2.6 years depending on the methodology and the latitude and altitude range considered (Patra et al., 2009; Waugh et al., 2013; Holzer and Waugh, 2015; Orbe et al., 2016, 2018; Yang et al., 2019; Orbe et al., 2021).

Based on $CH_4$ observations onboard the Greenhouse gases Observation SATellite (GOSAT) and simulations of an atmospheric chemistry-transport model (ACTM), Belikov et al. (2022) found that interhemispheric transport is most active in the altitude range of the upper troposphere, in a layer of 200 hPa below the tropopause. This agrees with in situ observations of a weaker gradient of $SF_6$ in the upper troposphere in comparison to the surface (Gloor et al., 2007). Using AoA as a diagnostic parameter, Krol et al. (2018) demonstrated in the Transcom AoA inter-comparison that interhemispheric exchange of air occurs faster in the upper troposphere (200-500 hPa) than near the surface. This results in locally steep vertical gradients of AoA. The parametrization of convection and thus vertical transport from the surface to higher altitudes was identified as an important factor for differences of interhemispheric transport time between models (Orbe et al., 2018; Krol et al., 2018).

While the interhemispheric gradient of $SF_6$ is mainly driven by the interhemispheric asymmetry of surface emissions, transport pathways from the surface to the upper troposphere also influence the latitudinal variation of mixing ratio at altitude (Miyazaki et al., 2009). In particular tropical convection can rapidly bring air masses with elevated mixing ratios of $SF_6$ and other tracers as for example from the Asian monsoon region or over tropical Africa to the upper troposphere (e. g. Randel and Park, 2006; Schuck et al., 2010; Vogel et al., 2016; Thorenz et al., 2017). Convection over remote marine tropical regions in contrast results in an inflow of air with low mixing ratios of anthropogenic tracers. This also implies that the interhemispheric gradient of $SF_6$ could vary with longitude as observed for example for $CH_4$ analysing measurements in the upper troposphere

from IAGOS-CARIBIC (In-Service Aircraft for a Global Observing System - Civil Aircraft for the Regular Investigation of the Atmosphere Based on an Instrument Container) and CONTRAIL (Comprehensive Observation Network for TRace gases by AIrLiner) flights into the southern hemisphere (Schuck et al., 2012).

Using an artificial age tracer in the Chemical Lagrangian Model of the Stratosphere (CLaMS), Yan et al. (2021) found that interhemispheric transport from the northern extra-tropics into the southern hemisphere occurs mainly in the altitude range 320–420 K, i. e. around the tropopause. They also confirmed that the Asian summer monsoon circulation interplaying with westerly ducts is a major driver of cross-hemispheric transport. The relevance of the monsoon circulation was also investigated in an idealized model study by Chen et al. (2017). They concluded that the zonally asymmetric heating associated with the Asian and North American monsoon has a significant impact on the AoA in the southern hemisphere. Without the monsoon circulation included in the model, the mean AoA since last contact with the northern extra-tropical surface becomes unrealistically large.

To assess such model results on interhemispheric transport and to constrain parametrizations of atmospheric transport processes, long-term measurements of long-lived tracers with pronounced interhemispheric gradients are needed. Such measurements should cover a large latitude range. Long time series of atmospheric trace gas observations are available from surface sites, and interhemispheric differences in tracer mixing ratios have successfully been used to constrain tracer emissions and budgets (Liang et al., 2014; Montzka et al., 2018) or have been interpreted as spatial shifts in emission patterns (Orbe et al., 2021). In the upper troposphere, in contrast, observational data are sparse.

Here, we use airborne observations in the upper troposphere covering the period 2006–2020 to investigate the interhemispheric gradient of $SF_6$. $SF_6$ is well suited to investigate interhemispheric transport processes, as its emissions exhibit weak seasonality with markedly different source strengths in the northern and the southern hemisphere. It is chemically inert and has no sinks in the troposphere or at the surface and can thus be considered an almost passive tracer of tropospheric transport processes. To a good approximation, it can be used for studies of tropospheric AoA as its emissions and mixing ratios are continuously increasing, although the increase has occurred faster than linear in recent decades (Rigby et al., 2010; Levin et al., 2010; Simmonds et al., 2020). Observational data are compared to output from the AGAGE 12-box model and the influence of transport parametrization on the observational-model agreement is investigated.

## 2  Data and Methods

### 2.1  Observational data

Interhemispheric gradients of $SF_6$ are investigated based on measurements from the IAGOS-CARIBIC project. IAGOS-CARIBIC is a long-term collaboration of several European scientific partners with the German airline Lufthansa. An instrument package inside an airfreight container was regularly deployed on board the A340-600 passenger aircraft *Leverkusen* (D-AIHE) during regular long-distance flights from December 2004 to March 2020. The aircraft was equipped with a sophisticated inlet system to allow measurements of trace gas and aerosol parameters in ambient air (Brenninkmeijer et al., 2007; Petzold et al., 2015).

The instrumentation included three air sampling units, two holding 14 glass flasks each, the third holding 88 stainless steel flasks (Schuck et al., 2009, 2012). Flights were over 2–4 consecutive days 6–12 times per year with usually a series of two to four long distance flights in series. Samples were collected at cruise altitudes of 400–175 hPa at predefined time intervals. Depending on ambient pressure, the total sample collection time is 30–90 s for glass samples and 70–240 s for stainless steel samples. To enhance the spatial resolution, sampling did not take place on each flight performed. Post-flight, the sampling units were removed from the instrument container for laboratory analysis. $SF_6$ mixing ratios were measured with gas chromatography (GC) coupled with an electron capture detector (ECD) (Schuck et al., 2009). $SF_6$ data are available for 7333 air samples covering the time period May 2006 to March 2020 with some gaps in the regular operation due to maintenance events. Spatial coverage can be inferred from Fig. 1 (gray symbols).

As part of the instrument package, carbon monoxide (CO) is measured using UV resonance fluorescence with a time resolution of 1 s (Scharffe et al., 2012), and ozone ($O_3$) is measured using UV photometry in combination with chemiluminescence (Zahn et al., 2012) with a time resolution of approx. 4 s. These CO and $O_3$ data are averaged over the sampling period for each individual canister sample.

CARIBIC data are supplemented by observational data from selected research aircraft missions. Missions were chosen such that data have significant latitudinal coverage and are representative of the upper tropospheric background, i. e., that the upper troposphere is well sampled and flights were conducted largely irrespective of special meteorological features. Based on these criteria, the missions TACTS (Transport and Composition in the UT/LMS, (Keber et al., 2020)) and SouthTRAC (Transport and Composition of the Southern Hemisphere UTLS, (Jesswein et al., 2021)) performed with the German High Altitude and LOng Range Research Aircraft (HALO) in 2012 and in 2019 were included (blue symbols in Fig. 1). TACTS took place out of Germany and flights covered a significant latitude span from the Cabo Verde islands to the Norwegian archipelago of Spitsbergen. SouthTRAC flights covered the central and southern Atlantic with transfer flights between Germany and southern Argentina via the Cabo Verde islands,

During both missions, $SF_6$ mixing ratios were measured with a time resolution of 1 min with the ECD-channel of the Gas chromatograph for Observational Studies using Tracers (GhOST) in situ instrument. $O_3$ measurements aboard HALO were performed based on UV-photometry with the Fast Airborne Ozone instrument (FAIRO), similar to the instrument used within the CARIBIC instrument package. During TACTS, CO and $N_2O$ were measured with a time resolution of 5 s with the TRIHOP instrument, a three-channel quantum cascade laser infrared absorption spectrometer (Müller et al., 2016). During the SouthTRAC mission, the University of Mainz Airborne Quantum Cascade Laser-spectrometer (UMAQS) instrument with a time resolution of 1 s was used (Müller et al., 2015; Kunkel et al., 2019).

In addition, data from the UCATS (Unmanned aircraft systems Chromatograph for Atmospheric Trace Species) instrument, (Hintsa et al., 2021), which operated during the aircraft missions HIPPO (HIAPER Pole-to-Pole Observations) and ATom (Atmospheric Tomography Mission) were included. During the HIPPO missions, which took place from 2009 to 2011 covering all seasons, the NSF/NCAR High-performance Instrumented Airborne Platform for Environmental Research (HIAPER) Gulfstream V aircraft was deployed to measure cross sections of trace gas mixing ratios over the Pacific and over North America (green symbols in Fig. 1), covering latitudes from 85° N to 65° S (Wofsy, 2011). The ATom mission was performed with the

**Table 1.** Overview of aircraft missions, instrumental precision and references for all data sets used.

| instrument and compound | detection method | precision | reference |
| --- | --- | --- | --- |
| **CARIBIC** | | | |
| whole air samples $SF_6$ | GC-ECD | 0.64 %[a] / 0.03 ppt | (Schuck et al., 2009) |
| whole air samples $N_2O$ | GC-ECD | 0.15 % / 0.2 ppb | (Schuck et al., 2009) |
| CO | UV-fluorescence | 1–2 ppb | (Scharffe et al., 2012) |
| $O_3$ | UV-photometry | 2 % | |
| **HALO missions** | | | |
| GhOST $SF_6$ | GC-ECD | 0.64 % | (Jesswein et al., 2021) |
| TRIHOP $N_2O$ | IR-absorption | 1.1 ppb | (Müller et al., 2016) |
| TRIHOP CO | IR-absorption | 1.0 ppb | (Müller et al., 2016) |
| UMAQS $N_2O$ | IR-absorption | < 0.5 ppb | (Müller et al., 2015; Kunkel et al., 2019) |
| UMAQS CO | IR-absorption | < 1 ppb | (Müller et al., 2015; Kunkel et al., 2019) |
| FAIRO $O_3$ | UV-photometry | 2 % | |
| **ATom/HIPPO missions** | | | |
| UCATS $SF_6$ | GC-ECD | 0.05 ppt | (Hintsa et al., 2021) |
| UCATS $N_2O$ | GC-ECD | 1.5 ppb | (Hintsa et al., 2021) |
| UCATS CO | GC-ECD | ~ 5 ppb | (Hintsa et al., 2021) |
| UCATS $O_3$ HIPPO | UV-photometry | 5 ppb[c] | (Hintsa et al., 2021) |
| UCATS $O_3$ ATom | UV-photometry | 2 ppb | (Hintsa et al., 2021) |

[a] precision improved in comparison to reference publication due to a modified peak integration algorithm

[b] precision value applies for upper troposphere data, different precision in low altitude marine data not used for this publication

NASA DC-8 aircraft over the years 2016–2018, also covering all seasons. Flights were over the Atlantic, over the Pacific, and over North America (red symbols in Fig. 1) covering latitudes from 83° N to 86° S (Thompson et al., 2022). Data from the latter two research aircraft missions thus extend the data set into the longitude range not covered by CARIBIC flights. The UCATS instrument combines two GC-ECD channels, one of which measures $N_2O$ and $SF_6$ every 70 s, while the other measures CO, among other gases, every 140 s. The instrument additionally includes UV absorption spectrometry measurements of $O_3$ at a time resolution of 10 s (Hintsa et al., 2021).

$SF_6$ from CARIBIC air samples and data from the UCATS instrument are reported on the WMO X2014 scale. Data measured prior to the publication of this scale were converted from the WMO 2006 scale to the WMO 2014 scale, both maintained by NOAA (https://gml.noaa.gov/ccl/sf6_scale.html). This was typically a correction of 0.01 ppt or less Data from the GhOST-ECD instrument are on the SIO-05 scale, and a conversion factor of $1.0049 \pm 0.002$ (WMO/SIO ratio) was applied to make absolute numbers comparable (Prinn et al., 2018). All $N_2O$ data are reported on the NOAA-2006A scale.

In addition to data from the above listed airborne instrumentation, data from the Global Monitoring Laboratory of the National Oceanic and Atmospheric Administration (NOAA) are used as reference data. Used are monthly mean values of the zonally-averaged Greenhouse Gas Marine Boundary Layer Reference of $SF_6$ (Lan et al., 2021) and monthly mean values of the $N_2O$ mixing ratio at Mauna Loa Observatory (MLO) (Dutton et al., 2017).

## 2.2 Selection criteria for tropospheric data

The presented data analysis combines several independent data sets that cover a large altitude range from the lower troposphere into the lowermost stratosphere. To select upper tropospheric data points from all observations, only measurements at altitudes with pressure $p < 400$ hPa are used. To exclude data from the lowermost stratosphere, data are filtered in several steps: (i) data are pre-filtered based on mixing ratios $\chi$ of CO and $O_3$ excluding measurements with $\chi_{CO} < 50$ ppb and $\chi_{O_3} > 120$ ppb, (ii) data points with an $N_2O$ mixing ratio of less than 97 % of the respective monthly mean value at MLO are not used because they are most likely stratospheric, (iii) an iterative baseline detection algorithm is used to identify low-$N_2O$ data and again tag them as stratospheric. The baseline detection is based on an iterative procedure fitting a time series function described by a second order polynomial combined with a first order harmonic (Schuck et al., 2018). Samples deviating from the baseline are sequentially removed from the data set until a further decrease of the standard deviation of the residual of the remaining baseline data set is less than 10 %. Removing very low values of $\chi_{N_2O}$ in the second filter step improves the stability of the baseline identification algorithm. The cut-off criteria in the pre-filtering steps are somewhat arbitrary but their exact values do not influence the results of the third step. Fig. 1 illustrates the measurement locations of all tropospheric observations above 400 hPa for which $SF_6$ measurements are available, excluding stratospheric data.

Figure 2 shows the resulting time series of $SF_6$ in the upper troposphere, excluding data flagged as stratospheric. Because of their higher time resolution, measurements from the UCATS and GhOST instruments exhibit a larger variability than CARIBIC canister samples which average over 30–240 s, depending on sampling unit type and altitude.

## 2.3 $SF_6$ lag time calculation

For the interpretation of the observed interhemispheric $SF_6$ gradient, the $SF_6$ lag time $\Gamma_{SF_6}$ is used. Following the definition by Waugh et al. (2013), the lag time $\Gamma_{SF_6}$ is the time offset fulfilling the condition that

$$\chi(x,t) = \chi_0(t - \Gamma), \tag{1}$$

with $\chi_0$ being a reference time series of the respective tracer, i. e. its source region mixing ratio, and $\chi(x,t)$ being the observed mixing ratio time series at location $x$ at time $t$. For the reference time series the zonally-averaged monthly mean values for 30–90° N observed at the marine boundary layer sites of the NOAA observational network were used. For each individual data point in the UT, the reference time series is approximated with a third order polynomial in a $\pm 10$ months window around the observation time and the lag time is calculated as the time offset, when the corresponding $SF_6$ mixing ratio is obtained.

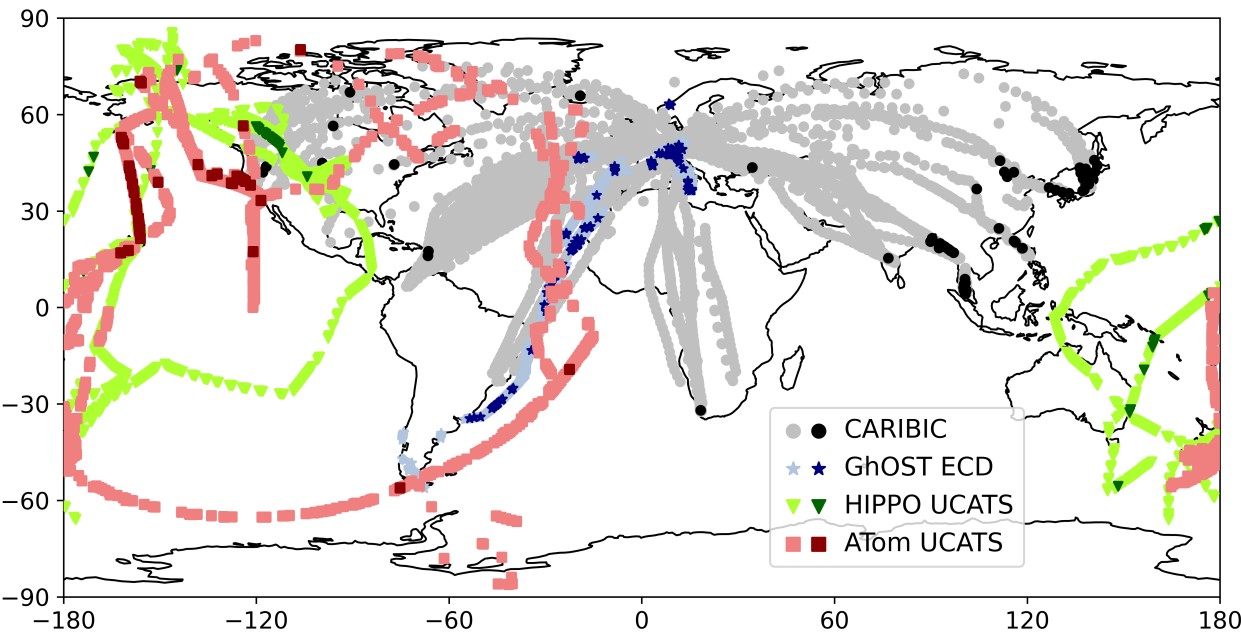

**Figure 1.** Map of all tropospheric measurement locations at altitudes above 400 hPa, excluding stratospheric data, colour-coded by data set. CARIBIC data are based on canister sampling with post-flight GC-ECD analysis, other data sets on in-flight measurements with GC-ECD instrumentation. Darker colours indicate samples with exceptionally high $SF_6$ mixing ratios (cf. section 3).

This approach removes the continuous increase in atmospheric mixing ratios from the data set and allows the combination of data from different observational periods in a consistent way. Note that the lag time depends on the increase rate and the interhemispheric difference in the emissions of the used tracer and therefore is different from a modelled AoA or transit time that quantifies the time since air was last in contact with the surface (Yang et al., 2019).

## 2.4  Box model

For simulations of interhemispheric lag times the Advanced Global Atmospheric Gases Experiment 12-box model is used, a two-dimensional model widely used for inversion estimates of global emissions of halogenated gases (Rigby et al., 2013). The model has three vertical layers: the lower troposphere (1000–500 hPa), the upper troposphere (500–200 hPa), and the stratosphere ($p < 200$ hPa), and four zonal bands: the southern extra-tropics (exT-S, 90–30° S), the southern tropics (T-S, 30–0° S), the northern tropics (T-N, 0–30° N), and the northern extra-tropics (exT-N, 30–90° N). For each altitude level, boxes are
numbered north to south: Boxes 0–3 in the lower troposphere, 4–7 in the upper troposphere and 8–11 in the stratosphere. The AGAGE 12-box model has recently been re-coded in Python (https://github.com/mrghg/py12box, last access Jan. 2023) with no conceptual changes to the previous version described in the supplementary material of Rigby et al. (2013).

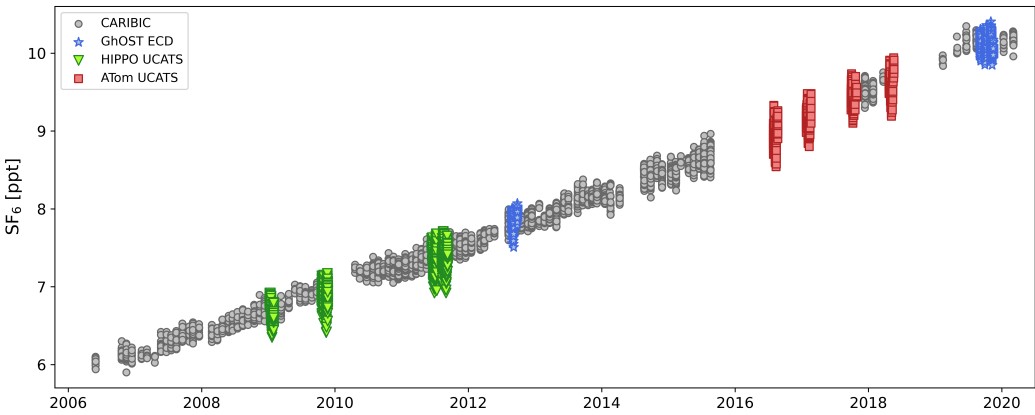

**Figure 2.** Time series of all upper tropospheric SF$_6$ mixing ratio observations. Gray symbols represent CARIBIC flask sampling data, coloured symbols represent in-flight measurements from aboard research aircraft. These high frequency data show higher variability of the data. Stratospheric measurements were excluded based on simultaneous measurements of CO, O$_3$ and N$_2$O. The black line indicates the monthly mean mixing ratios in the northern hemispheric marine boundary layer used as reference time series.

For species with tropospheric losses through reaction with OH, which is not the case for SF$_6$, the model uses the annually repeating OH field by Spivakovsky et al. (2000). Stratosphere-troposphere exchange is based on a single mixing time-scale,
and stratospheric loss is parametrised by a seasonally varying loss rate in each of the stratospheric boxes. Transport between the individual boxes is realised by parametrising bulk advection and eddy diffusion with the latter term dominating the transport (Cunnold et al., 1983, 1994). This is done with a transport matrix $T$, with elements $T_{ij}$, that quantify transport between pairs of individual boxes. These transport parameters vary seasonally but have no inter-annual variability. They were derived based on the best fit to chlorofluorocarbon measurements from the AGAGE observational network and can be assumed to yield
reasonable results for gases with similar emission characteristics, which holds for gases emitted mainly from anthropogenic sources such as SF$_6$ (Cunnold et al., 1983, 1997, 2002).

The model is run for the period 1990–2020 using zonally averaged SF$_6$ emission fluxes from the Emissions Database for Global Atmospheric Research (EDGARv7.0) (Crippa et al., 2021) assuming monthly emissions of 1/12 of the annual total. For initialization, mixing ratios of 2.67 ppt and 2.34 ppt are used for the northern and southern hemisphere extra-tropics for the
185 year 1990 (Maiss et al., 1996). The northern tropics are initialized with a mixing ratio of 2.53 ppt applying a 0.14 ppt offset to the extra-tropics (Maiss et al., 1996), the southern tropics were interpolated to a mixing ratio of 2.44 ppt.

Upper tropospheric boxes are initialized with identical values to the surface boxes. The four stratosphere boxes are initialized with a 0.2 ppt offset, i. e. with mixing ratios lagging behind the upper troposphere. This value was derived as the average offset of the lowermost stratosphere with regard to the upper troposphere from the cross-tropopause gradient of the CARIBIC data
set for the northern hemisphere mid-latitudes using data with a potential temperature difference of 5 K above the thermal tropopause. While absolute values of the modelled SF$_6$ mixing ratios crucially depend on these initial conditions, this potential

bias is removed when discussing SF$_6$ time lags rather than mixing ratios. The northern hemisphere lower tropospheric extra-tropical box is used as the reference time series for the time lag calculations. Model results for the four stratospheric boxes are not evaluated here.

## 3 Interhemispheric gradient of SF$_6$ in the upper troposphere

Exemplary for the interhemispheric gradient of SF$_6$ mixing ratios in the upper troposphere, Fig. 3 shows the latitudinal variation of SF$_6$ mixing ratios for flasks collected during CARIBIC flights into the southern hemisphere. These flights took place between March 2009 and March 2020 and continuously increasing mixing ratios are observed. While during flights over the Atlantic to South America (triangles in Fig. 3), a continuous decrease of mixing ratios with latitude is observed, flights over land to South Africa sometimes show elevated mixing ratios in the equatorial regions (e. g. top-most profiles in Fig. 3), similar to what was observed for other gases (Schuck et al., 2012; Thorenz et al., 2017). Thus, the latitudinal gradient also has a longitudinal variation depending on emission and transport patterns.

Because some monthly latitudinal mixing ratio profiles show an elevation in the tropics, a simple linear fit does not describe the profiles well, and we use the interhemispheric difference as a metric to compare observations outside $\pm 30°$. To derive this value, monthly means were calculated for the extra-tropics of each hemisphere, and the difference of the zonal means was taken for each month with observations in both hemispheres available. Because the mixing ratio growth rate is almost constant, differences are averaged for all observations shown in Fig. 1, resulting in an average interhemispheric difference of 0.19 ppt. At the ground, using zonal differences north and south of $\pm 30°$ from the Greenhouse Gas Marine Boundary Layer Reference of SF$_6$ (Lan et al., 2021), indicated by horizontal lines in Fig. 3, an average difference of 0.33 ppt is derived for the period May 2006 to March 2020 which corresponds to all upper tropospheric observations. Using an earlier CARIBIC flight from Germany to Cape Town performed in December 2000 (data are not part of the data set analysed here), Gloor et al. (2007) derived an interhemispheric difference of SF$_6$ of 0.15 ppt, again smaller than what was observed at the surface (0.38 ppt). This is consistent with the assumption that interhemispheric transport is most active in the upper troposphere (Belikov et al., 2022).

The continuous increase of atmospheric SF$_6$ mixing ratios complicates the analysis of data covering more than a decade. Therefore, the SF$_6$ lag time according to equation 1 is used in the following. The data analysis combines several data sets with large differences in coverage and resolution in space and time. Fig. 4 (a) shows the resulting time lags as a function of latitude colour-coded by aircraft mission. Shown are individual data points (dots) and zonal averages over 5° latitude (symbols), excluding latitude bins with less than five observations. All four data sets agree with each other within their respective variability. In particular, this is the case for data from the HIPPO flight which cover a longitude range that is under-represented in the CARIBIC data and not covered by the included HALO missions. From this it can be concluded that the longitudinal variability is smaller than the interhemispheric gradient. Thus, all observations are combined applying the above filter procedure to all observations simultaneously and zonal averages are discussed in the following.

In Fig. 4 (b), the resulting latitudinal profile of SF$_6$ time lags in the upper troposphere is shown. In the northern mid-latitudes and sub-tropics, time lags scatter around 0, with negative values of individual samples up to -2.4 years and positive values up to

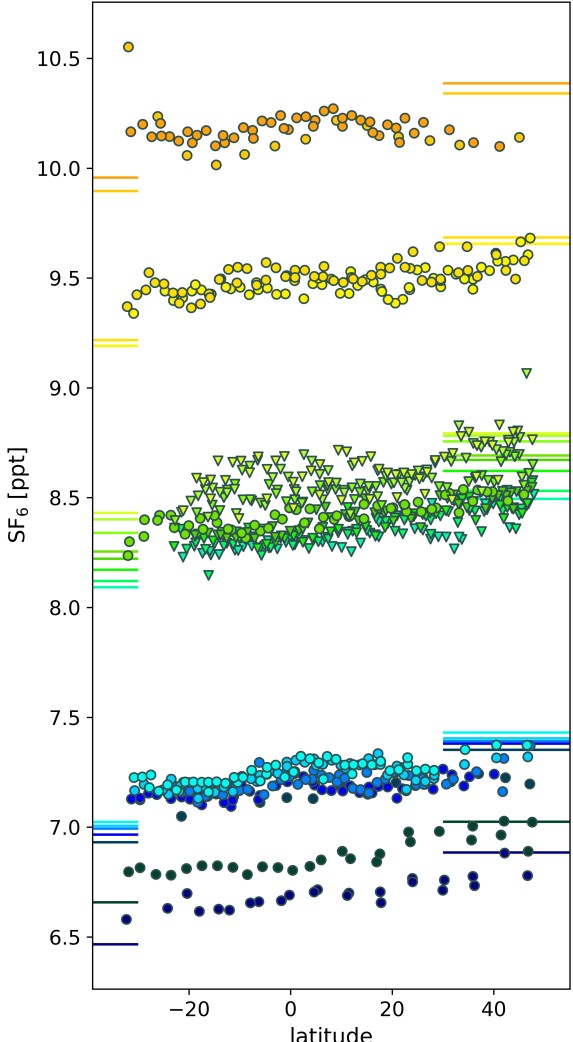

**Figure 3.** Latitudinal profiles of SF$_6$ mixing ratios measured in CARIBIC flask samples collected during flights into the southern hemisphere including high SF$_6$ extreme values. Flights took place between May 2009 and March 2020, with flights between Germany and South Africa represented by circles and flights to South America by triangles. Horizontal lines denote surface mixing ratios for the extra-tropical latitudes outside $\pm 30°$ of each hemisphere. Different colours are used to group flights by year and month to guide the eye only, therefore a legend is omitted.

2.2 years. While large positive numbers reflect stratospheric influence on upper tropospheric mixing ratios, negative lag times are likely related to recent rapid uplift of anthropogenically influenced air with mixing ratios significantly above the marine boundary layer background chosen as reference. For their identification, the iterative baseline identification algorithm (cf. step (iii) in section 2.2) was applied to the time series of SF$_6$ mixing ratios to tag extreme events with lag times significantly below

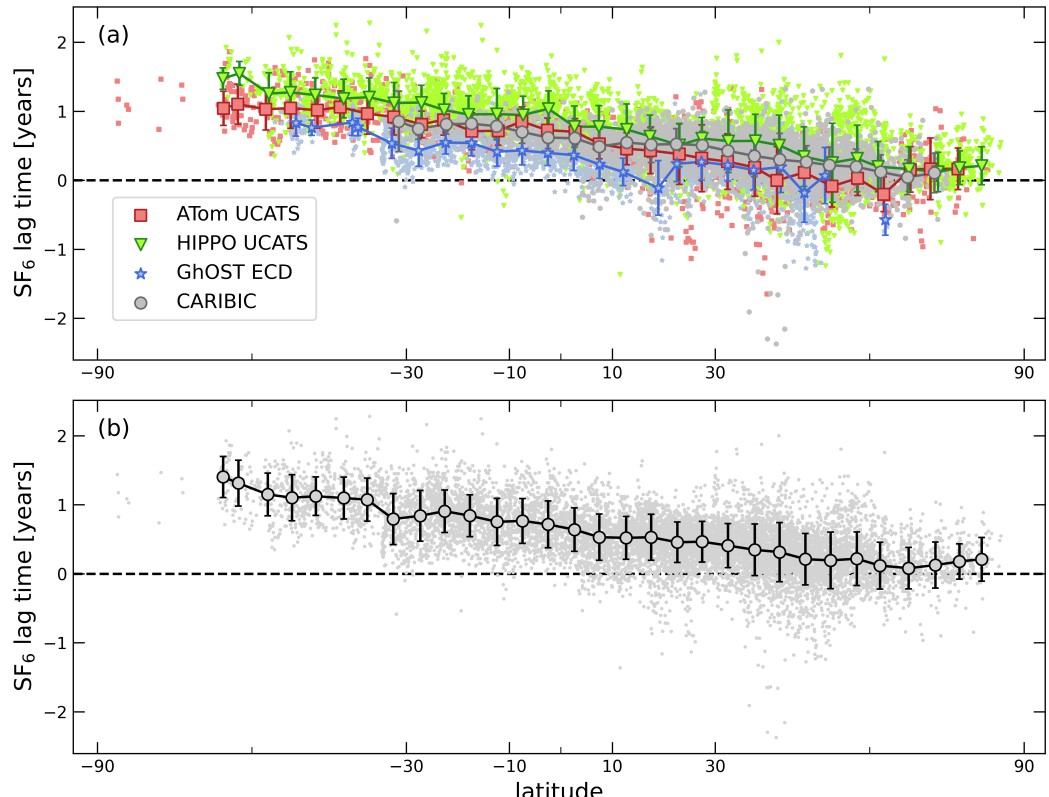

**Figure 4.** Observation-based time lags as a function of latitude excluding data that were identified as stratospheric. Dots represent individual SF$_6$ measurements, symbols are 5-degree zonal averages with the standard deviation of each interval indicated by the error bar. Latitude bins containing less than five data points are not shown. Panel (a) differentiates data by instrument and mission, panel (b) shows the combined data set.

the baseline. These are highlighted by darker symbols in the map in Fig. 1 and mainly occur in the mid-latitudes of the northern
hemisphere. The spatial distribution of lag-times is shown in the map in the supplementary Fig S1.

South of 10° S, lag times become continuously larger towards more southern latitudes reaching values around 1 year around 60° S where data coverage becomes sparse. Compared to the meridional profile derived from ground observation sites of the NOAA network by Orbe et al. (2021), who reported time lags of 1.5 years at the surface in the southern extra-tropics, the gradient across the tropics in the upper troposphere is not as steep and time lags in the southern hemisphere upper troposphere
are smaller than in the boundary layer. Again, this is consistent with previous observations of a smaller interhemispheric difference in the upper troposphere in comparison to the surface (Gloor et al., 2007). Despite the weaker gradient, transport barriers are reflected in the data by slight changes in the slope of the latitudinal profile of lag times around 10° and 30° of each hemisphere.

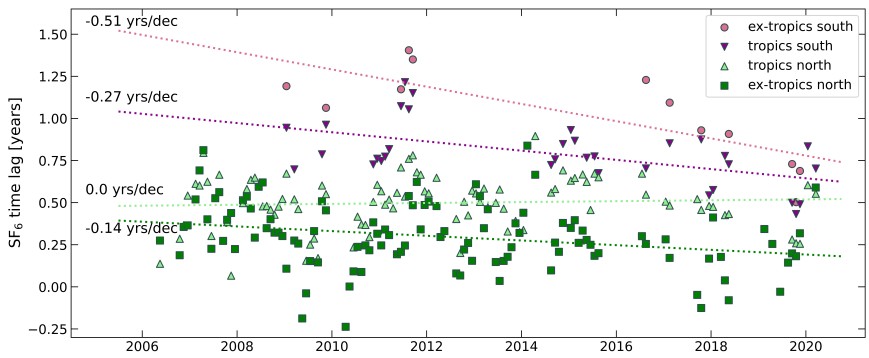

**Figure 5.** Time series of monthly mean values of the $SF_6$ time lag with respect to the northern extra-tropical boundary layer background in the upper troposphere in four zonal bands. Lines represent results of linear regressions to each zonal band.

Fig. 5 shows monthly mean values of the observed time lags as a function of time for the four zonal bands defined in the 12-box model. For the calculation of monthly means, large negative $SF_6$ lag times are excluded applying the statistical baseline detection described above to obtain results representative for the upper tropospheric background. Also shown are the results of a linear regression for each zonal band. In the northern hemisphere tropics no statistically significant ($2\sigma$) trend is observed, whereas in the southern tropics a trend of -0.27±0.01 years/decade is derived. Trends for the northern and southern hemisphere extra-tropics are -0.14±0.05 years/decade and -0.51±0.13 years/decade, however, the number of observations in the southern extra-tropics is small and seasons are not equally represented. The error values of the trends are statistical errors resulting from the fitting procedure, possible systematic errors due to inhomogeneous data coverage or the different instrument time resolutions are not included. A decreasing time lag in the southern hemisphere is consistent with earlier findings of $\sim$ -0.12 years/decade based on surface data, but is in contrast to smaller trends at the surface in the northern tropics and a positive trend in the northern extra-tropics (Orbe et al., 2021). Trends derived from surface data were found to strongly depend on the selection of observational sites and the choice of the reference time series. Orbe et al. (2021) also included data from the ATom mission in their analysis, combining results from the UCATS instrument with flask sample analysis. In their vertical profiles they derived larger $SF_6$ time lags at altitudes between 400 hPa and 200 hPa than shown here, one possible reason for this is that stratospheric data were included in their analysis. Small differences may also arise from a different choice of the reference time series.

## 4 Comparison of observations with model results

Changes in the $SF_6$ time lag over time could be related to a change in emission patterns or to a change in interhemispheric transport. Both would modify the relation between the reference time series and the time series observed in the upper troposphere at distance to where the emissions occur. Comparisons with other trace gas gradients could provide additional information, but observations of suitable tracers in the upper troposphere with sufficient spatial and temporal coverage are sparse. The

ideal tracer for such studies would be long-lived compared to atmospheric transport times, have a constant emission difference between the hemispheres, no seasonality in its emissions and no tropospheric or surface sinks. In reality, most industrial tracers have varying spatial emission patterns with less well defined interhemispheric gradients, shorter lifetimes, and they are removed from the atmosphere through reaction with the OH radical already in the troposphere or are not measurable with sufficient precision due to their low atmospheric mixing ratios.

Therefore, to investigate the influences of emission fluxes and interhemispheric transport on the interhemispheric gradient of $SF_6$, the AGAGE 12-box model is used in the following. We first compare the modelled interhemispheric mixing ratio difference to the observed one. From the observations in the upper troposphere, we found an average north-south difference of 0.19 ppt, whereas at the surface it was 0.33 ppt. Calculating the same numbers from the model output for the period 2006–2020, 0.51 ppt and 0.57 ppt are obtained. In agreement with the observations, the interhemispheric mixing ratio difference of

$SF_6$ is larger at the surface than in the upper troposphere, however, in the model the gradient is much larger. This points to either the interhemispheric difference of the emission scenario used for the model calculation to be too large or a too strong interhemispheric transport barrier between the northern and the southern hemisphere in the model transport scheme. In addition, the difference between the surface and the upper troposphere is smaller in the model than observed (cf. Fig. S5).

Absolute mixing ratio numbers obtained from the AGAGE 12-box model depend on the initialization values, therefore a

relative quantity such as the time lag is more appropriate for comparison of observations and model output. For evaluation of the box model output, the northern extra-tropical lower troposphere box (Box 0 - PBL exT-N) is used as the reference time series. The comparison of the modelled and the observed time lag is shown in Fig. 6, and a comparison of absolute mixing ratio values is included in the Supplement (Fig. S2). While good overall agreement is seen in the northern hemisphere extra-tropics, observed time lags (coloured lines and symbols) are on average slightly smaller in the northern tropics and much

smaller than model output in the lower and upper troposphere in the southern hemisphere (black lines). Fitting a trendline in each latitude band over the time period covered by observations as visible in Fig. 5, modelled trends are negative in all four upper tropospheric boxes, but are not statistically significant ($2\sigma$). Only in the northern extra-tropics (Box 4), a statistically significant trend of -0.07±0.02 years/decade is obtained, smaller than the value of -0.14±0.05 years/decade in the observations.

Similar results were obtained previously using the more sophisticated NASA Global Modeling Initiative chemical trans-

port model (CTM) (Strahan et al., 2007, 2016). Waugh et al. (2013) compared the CTM output to ground-based, ship-borne, and aircraft measurements from the NOAA observational network and found the model to overestimate lag-times towards southern latitudes. At middle and high latitudes in the southern hemisphere, ground station observations yielded lag-times of 1.3–1.4 years, whereas the CTM results were around 1.75 years for latitudes south of 30° S. Analysing transit time distributions derived from CTM results, Orbe et al. (2016) obtained modelled mean tropospheric age values of 1.5–2 years from the surface

up to 200 hPa in the southern extra-tropics. Comparing results of a newer model run to surface observations, Orbe et al. (2021) reported good agreement between surface observations and the model results in the northern hemisphere, but an increasing overestimation by the model towards southern latitudes. In the southern extra-tropics observation-based lag-times of approximately 1.5 years were significantly below the model result of approximately 2 years. The overestimation was largely attributed

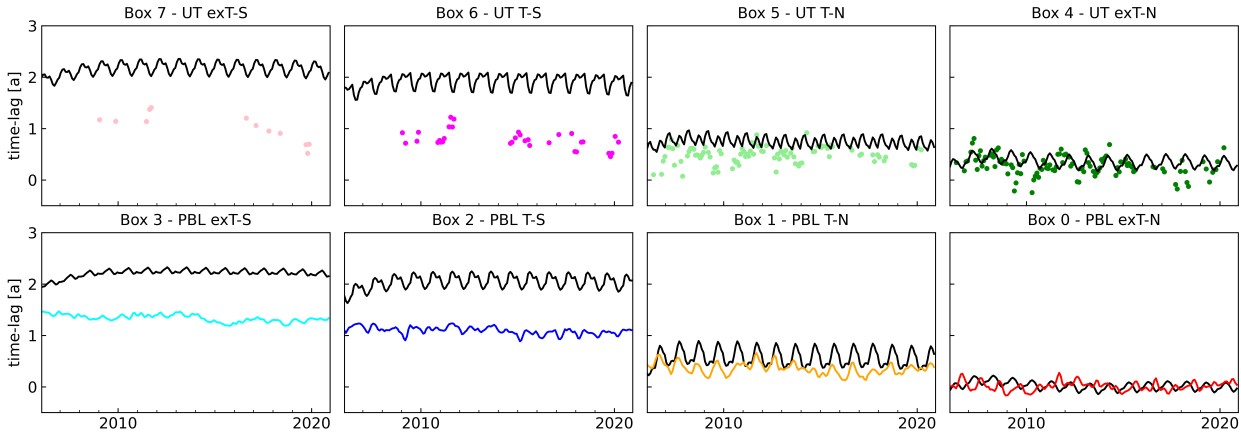

**Figure 6.** Comparison of time lags from the AGAGE 12-box model (black lines) with upper tropospheric aircraft observations (Boxes 4, 5, 6, 7) and measurements in the planetary boundary layer (Boxes 0, 1, 2, 3). Upper troposphere observations do not include exceptionally high mixing ratios of $SF_6$.

to the influence of high-$SF_6$ sites on the reference time series used to calculate the model lag times. This supports our choice
of using the marine boundary layer zonal average as the reference time series.

Reasons for systematic deviations of time lags in the model could be related to emission strength or the spatial distribution of emissions or to the transport parameters used in the model. To assess these factors, several sensitivity runs were performed which are summarized in Table 2. To quantify the model-observation difference and to identify the parameter setting that agrees best, the mean absolute deviation ($\mathrm{MAD}_{\mathrm{Box}\,i} = 1/N \cdot \sum_{j=0}^{N} |x_{\mathrm{model},j} - x_{\mathrm{obs},j}|$) index is calculated over all observations
$x_{\mathrm{obs}}$ and corresponding model output $x_{\mathrm{model}}$ for each model box. The index $j$ counts the monthly timesteps of the model runs. $\mathrm{MAD}_{\mathrm{Box}\,i}$ can be calculated independently for mixing ratios and time lags. For the scaling of global emissions the MAD calculation is done on the basis of mixing ratio values because the global scaling does not affect the time lag differences between the observations and the model. For the sensitivity experiments optimizing the transport scheme the time lag parameter is used. In general, we find that an optimisation of the difference between model and observations based on mixing ratios also
results in an improvement of the time lag difference but not always vice versa. To quantify model-observation agreement with one single parameter, the MAD values of the eight boxes are averaged by calculating the Euclidean distance $\mathrm{d(MAD)} = \sqrt{(\sum_{i=0}^{7} \mathrm{MAD}_{\mathrm{Box}\,i}^2)}$. This gives more weight to an improvement in boxes with initially poor agreement. For a particular setting, the best model setup is chosen as that with the minimum d(MAD) value.

First, as the modelled mixing ratios are consistently below the observations in all tropospheric boxes over the complete
observation period, EDGAR v7.0 emissions were scaled globally with a factor 0.9–1.1 in steps of 0.00025. The lowest overall MAD is obtained for an upscaling of the emissions by 3.25 %. An underestimation of emissions by the EDGAR inventory would be in agreement with earlier findings referring to earlier versions of the inventory (Simmonds et al., 2020; Rigby et al., 2010). As Fig. 6 indicates that in particular emissions at southern latitudes might be too low, up to 28 % of emissions were

**Table 2.** Overview of sensitivity experiments performed with the box model. The ranges given are the maximum variation of emissions $E_i$ into the boundary layer boxes or transport parameters $T$. The best estimate is determined from the smallest mean absolute deviation of the mixing ratios (time lag for transport only experiments) averaged over the eight tropospheric boxes. If no best estimate is listed in the third columns, no reasonable minimum was found. The respective best parameter estimates are finally used to limit the range of the multi-parameter model optimisation experiment. The best parameter scalings estimated in the multi-parameter approach are listed in the fourth column.

| parameter varied | range | best estimate single exp. | best estimate combined exp. |
|---|---|---|---|
| global scaling of emissions | | | |
| $E_{total}$ | [0.95; 1.1] | 1.0325 | |
| zonal scaling emissions | | | |
| $E_0$ | [0.7; 1.15] | 0.72 | 0.75 |
| $E_1$ | [0.9; 4] | 1.8 | 1.75 |
| $E_2$ | [0.9; 4] | – | 1.70 |
| $E_3$ | [1.2; 2] | – | 1.70 |
| modified interhemispheric transport | | | |
| PBL: $T_{12}$ | [0.07; 1] | 0.34 | |
| UT: $T_{56}$ | [0.07; 1] | – | |
| modified transport | | | |
| tropics PBL: $T_{12}$ | [0.1; 0.8] | 0.83 | 0.30 |
| tropics UT: $T_{56}$ | [0.05; 1] | 0.15 | 0.30 |
| NH: $T_{01}$ and $T_{45}$ | [0.1; 1] | 0.47 | 0.70 |
| SH: $T_{23}$ and $T_{67}$ | [0.1; 2] | 1.19 | 1.20 |

taken out of the northern hemisphere extra-tropics (Box 0) and shifted southward into the tropics (Boxes 1 and 2). This would

be consistent with findings by Hu et al. (2023) that $SF_6$ emissions from the United States are overestimated by the EDGAR inventory. Following a set of intermediate runs performed in coarse steps to narrow the scaling intervals first, emissions into each boundary layer box were scaled independently over the ranges listed in Table 2. Note that the range column lists the extreme limits, but not each experiment covered the full range. The best result of this sensitivity run is shown in blue (triangles and dotted lines) in Fig. 7 and 8. The value of d(MAD)$_{mxr}$ improved from 0.66 ppt in the reference run with default model

settings to 0.34 ppt.

For the boxes in the southern hemisphere (Boxes 2 and 3), "optimized" emissions were always found at the maximum range allowed for scaling, even if the scaling interval was increased to extreme values up to a factor of 4. This behaviour indicates that additional emissions may compensate for other effects such as the transport scheme. For an investigation of the model

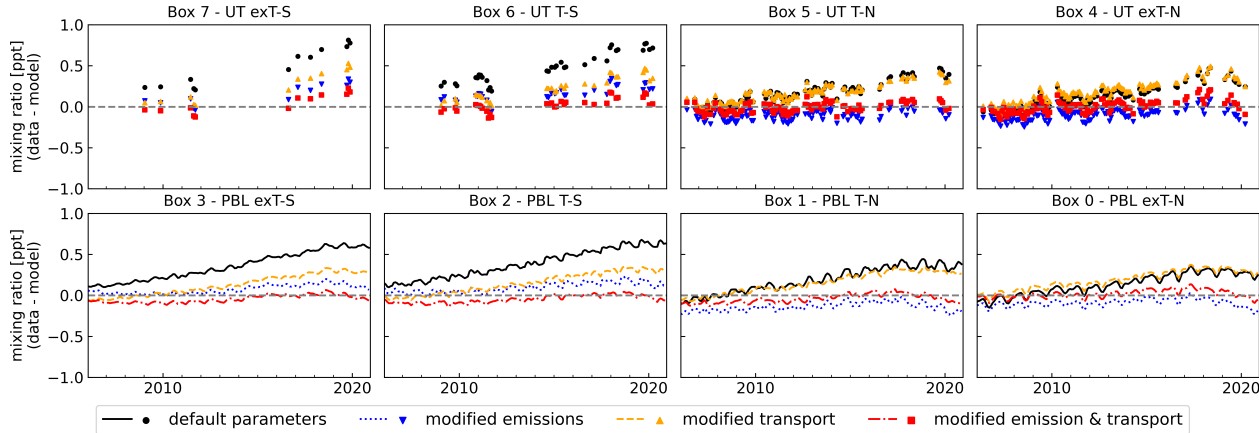

**Figure 7.** Volume mixing ratio difference between observations and selected model experiments. Black circles and line represent the default model setup, blue triangles and line the modified emissions run, orange triangles and lines the modified transport run and red squares and lines the combined model experiment. Upper troposphere observational data do not include samples with exceptionally high mixing ratios of $SF_6$.

transport parameters, we first evaluate the time lag difference between neighbouring boxes in the horizontal and the vertical. Results are plotted in Fig. S5 and S6 of the Supplement. While deviations are small in the vertical, the largest discrepancies are found between the northern and the southern tropics in the lower and in the upper troposphere (Boxes 1 and 2 (transport parameter $T_{12}$) and Boxes 5 and 6 ($T_{56}$)). We therefore next vary only these two parameters independently. Since the transport is dominated by eddy diffusion (Cunnold et al., 1983), variation of transport in the model is achieved by scaling the eddy diffusion parameters, with the advection parametrisation kept unchanged. Note that the matrix elements $T_{ij}$ listed in Table 2 are used inversely, thus a scaling with values $< 1$ represents faster transport. Furthermore, the scaling of transport parameters was done equally in all months, while the parameters themselves change monthly. For $T_{56}$ no minimum is found, and transport across the equator in the upper troposphere (Boxes 5 and 6) becomes instantaneous, indicating that it is not sufficient to only vary these two transport parameters. In a fourth experiment, interhemispheric transport ($T_{12}$ and $T_{56}$) and transport between the tropics and the extra-tropics of each hemisphere ($T_{01}$ and $T_{45}$ in the NH, $T_{23}$ and $T_{67}$ in the SH) were varied. The best result of this sensitivity experiment is shown in orange (triangles and dashed lines) in Fig. 7 and 8.

Finally, the emission varying experiments and the transport variation are combined using the results of the first simple sensitivity runs to reduce the computational effort by narrowing the parameter intervals. The resulting mixing ratio and lag time series comparisons for an optimised emission scheme and an optimised transport scheme are shown in red (squares and dash dotted lines) in Fig.7 and 8. The value of d(MAD)$_{mxr}$ improves to 0.18 ppt for this model setup, evaluating the lag time differences between model and observation d(MAD)$_{lag}$ improves to 0.47 years compared to 1.95 years with the default transport scheme and unscaled emissions. Column *best estimate single exp.* of Table 2 lists the resulting best value of the respective parameters(s) relative to the default setting.

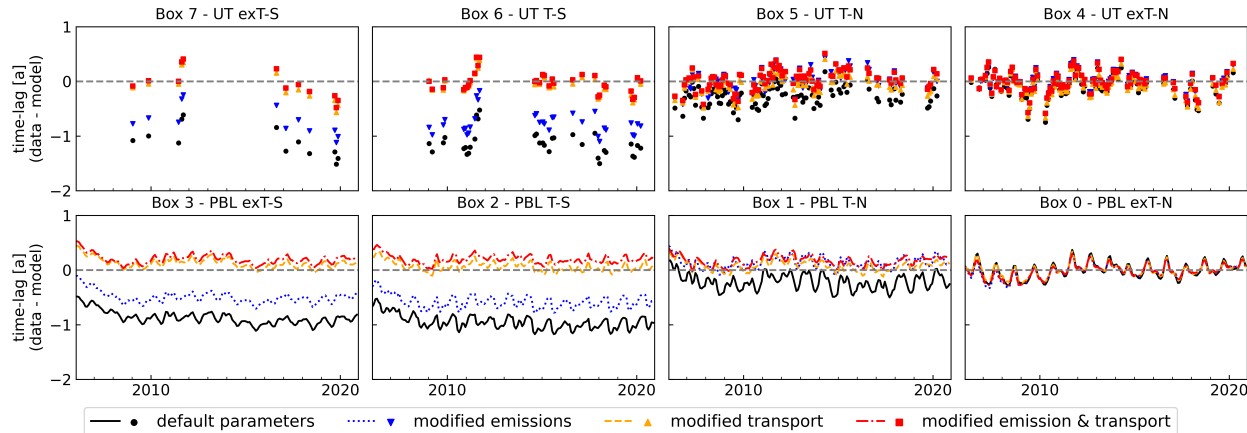

**Figure 8.** As Fig. 7, showing time lag differences.

## 5 Conclusions

We used upper tropospheric airborne in situ observations of $SF_6$ to investigate the interhemispheric gradient of this tracer and

to assess interhemispheric transport in the AGAGE two-dimensional 12-box model (Rigby et al., 2013). Upper tropospheric observational data cover a latitude range from $\sim 80°$ N to $\sim 60°$ S for the period 2006–2020. Observations are attributed to the troposphere using an $N_2O$-based statistical filter in combination with mixing ratios of CO and $O_3$.

The analysis shows a weaker interhemispheric mixing ratio difference of 0.19 ppt in the upper troposphere compared to observations in the marine boundary layer which yield 0.33 ppt. The marine boundary layer time series in the northern extra-

tropics is used as a reference to derive interhemispheric lag times. With little vertical gradient in the northern extra-tropics, north of $30°$ N, lag times around 0 years are observed also in the upper troposphere. But also some individual extreme events with negative lag times up to -2 years are apparent which are indicative of recent transport of industrially influenced air masses from the surface. These observations of extremely high $SF_6$ mixing ratios are identified with a statistical approach and are excluded from the later model comparison.

Across the tropics, lag times in the upper troposphere increase southward and reach values higher than 1 year south of $30°$ S. The observations are evaluated in four zonal bands ($>30°$ S, $30°$ S–$0°$, $0°$–$30°$ N, $>30°$ N) following the latitude limits of the 12-box model used for comparison. In the northern tropical upper troposphere no trend of the derived lag time is found over the observation period, but lag times decrease with time in the southern tropics and the extra-tropics. This does not necessarily imply a change in interhemispheric transport, but could also reflect a change in the spatial emission patterns with a weaker

emission increase in the northern extra-tropics compared to the tropics and southern extra-tropics.

For comparison with the observations, the AGAGE two-dimensional 12-box model is used (Rigby et al., 2013). The model has two tropospheric altitude levels defined as $p > 500\,hPa$ and $200\,hPa > p > 500\,hPa$. Using zonally averaged bottom-up emission estimates from EDGAR 7 data, modelled $SF_6$ mixing ratios are systematically too low. In the northern extra-tropics

at lower and higher altitudes, a mean underestimation of 0.14 ppt and 0.12 ppt is found that increases to 0.36 ppt in the southern
extra-tropics at lower altitudes and 0.50 ppt at higher altitudes. Comparing lag time differences derived analogously from model
output and the observations, the model-observation difference is only -0.01 years near the northern extra-tropical surface, but
an overestimation of the time lag of -1.1 years is obtained for the southern hemisphere upper troposphere. Comparing the
model-observation difference for the eight tropospheric boxes of the model, horizontal transport seems to be more important
than vertical transport.

To study the influence of the model transport scheme, a series of sensitivity runs was performed in which better agreement
was found with a 3.25 % global emission increase relative to EDGAR 7 bottom-up emissions, a southward shift of emissions
and a modified transport scheme with a weaker tropical transport barrier and thus faster transport into the southern hemisphere.
In particular, agreement between the lag time in the model and the observation-derived lags improves with faster transport
from the northern hemisphere extra-tropics into the tropics and across the equator. This agrees with earlier findings that the $SF_6$
time lag is very sensitive to transport from the northern extra-tropics into the tropics (Yang et al., 2019). In contrast, transport
from the southern hemisphere tropics into the southern extra-tropics is too fast in the 12-box model, and better agreement is
obtained with slower transport within the southern hemisphere.

The simple setup of the two-dimensional box model was chosen to test a large set of parameters and to investigate the poten-
tial of upper tropospheric observations as an additional constraint for estimates of global emissions. It does not allow to clearly
disentangle the relative contributions of emissions and the transport scheme. To address this open issue, a more sophisticated
model with artificial age tracers implemented would be needed. Another aspect, that has not been considered here, is transport
across the tropopause. As previously shown, seasonally varying transport across the extra-tropical tropopause is a relevant
factor for the derivation of stratospheric age of air, in particular near the tropopause (Hauck et al., 2020; Wagenhäuser et al.,
2023). Cross-tropopause transport will also influence upper tropospheric trace gas mixing ratios and transit times. Despite these
limitations, our analysis shows the potential of upper tropospheric trace gas observations to constrain atmospheric transport
processes and to provide additional constraints for inverse modelling of surface emissions.

*Code availability.* Model code is available on https://doi.org/10.5281/zenodo.6857447, see GitHub https://github.com/mrghg/py12box for
most recent version.

*Data availability.* The dataset of $SF_6$ mixing ratios in the upper troposphere used for this publication is available via https://zenodo.org/
records/10018398 CARIBIC flask sampling data can be requested from the project coordinators via https://www.caribic-atmospheric.com/
Data.php. Observational data from the HALO missions are available via the HALO Database (https://halo-db.pa.op.dlr.de/). ATom observa-
tional data are available at the Oak Ridge National Laboratory Distributed Active Archive Center (ORNL DAAC; (https://doi.org/10.3334/
ORNLDAAC/1925) (Wofsy et al., 2017). HIPPO observational data are available at the Earth Observing Laboratory data archive (EOL data
archive, https://doi.org/10.3334/CDIAC/HIPPO_012) (Wofsy et al., 2021).

*Author contributions.* TJS lead the data analysis, drafted the manuscript and performed parts of the CARIBIC flask sample analysis. JD performed data analysis, carried out the box model calculations and prepared figures for the manuscript. TJS, AE, MJ, TW, TK, PH, DK, EH, FM, FO, and AZ contributed observational data and participated in aircraft missions. MR and LW coded the current box model version and gave advice for the calculations. All co-authors were involved in the scientific discussion and editing of the article.

*Competing interests.* TJS and AE are members of the editorial board of *Atmospheric Chemistry and Physics*.

*Acknowledgements.* The authors acknowledge the contribution of all staff who performed regular maintenance of the IAGOS-CARIBIC container, flight preparation and handling of the air sampling units, in particular Torsten Gehrlein, Claus Koeppel, Dieter Scharffe and Stefan Weber. Also the contribution of all technical staff during research aircraft campaigns is appreciated. We also acknowledge NOAA Global Monitoring Laboratory for providing surface $SF_6$ measurements.

*Financial support.* This research has been supported by the Deutsche Forschungsgemeinschaft (grant nos. SCHU 3258/1-1 - Project-ID 401669047, SCHU 3258/3-1 - Project-ID 501095243, and DFG collaborative research program "The Tropopause Region in a Changing Atmosphere" TRR 301 – Project-ID 428312742).

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
