# Peer review of "The interhemispheric gradient of $SF_6$ in the upper troposphere"

_EGUsphere, 2023_

## Author Comment (AC1)

**Schuck et al., The interhemispheric gradient of SF6 in the upper troposphere**

**Response to comments by Anonymous Referee #1**

We thank the anonymous reviewer #1 for the thorough reading of the manuscript and their suggested improvements. All comments are addressed in the following with the reviewer's comments printed in blue, and the responses in black.

This article analysed long-term measurements of SF6 from many aircraft campaigns, covering a period of 2006-2020. They have calculate the age of air based on the aircraft observations in the upper troposphere, in reference to continuous surface measurements. Further explanation to the observed results are derived from 12-box model of the atmospheric transport. The paper is very well written and organised.

My main concern is the use of a very low resolution of the transport model, which I think would be very difficult convince the reader as a state-of-the-art. The limitation of 12-box model are known and I do not need to elaborate (overstretched to compare with measurements). Given the importance of the data set prepared for this analysis and discussion of the possible outcome, I would recommend publication of the article after minor revision. Hopping that the results presented will inspire future research activity.

The 12-box model was chosen as a very simple setup, which allowed us to test a large set of parameters. One purpose of the study was to investigate the potential of upper tropospheric observations as an additional constraint for estimates of global emissions. The limited 12-box model enabled us to investigate the influence of emissions and transport parameters on the model output, which would not have been possible to such an extent with a complex model driven by meteorological data and parametrizations of individual transport processes.

Minor comments:

Line 21 : Aren't SF6 lost also " by electron attachement" ?

Electron attachment has been identified as a relevant loss process of atmospheric SF6, occurring in the mesosphere. This fact has been added to the manuscript text including Kovács et al. 2017 as an additional reference.

Line 24 : I think the age of air concept was laid out first in Bischof et al., 1985

Bischof et al 1985 and Hall et al. 1998 were added to the selected references cited in this part. The latter was chosen as one of the very first publications discussing SF6 stratospheric mean age.

Line 38 : This phenomena was first reported elsewhere, using CO2 in a landmark paper (Nakazawa et al., 1991).

We agree that Nakazawa et al. 1991 report similar observations for CO2, comparing the interhemispheric difference in the upper troposphere and lowermost stratosphere to results of ground-based measurements. Owing to the strong seasonal cycle, the interhemispheric and

vertical gradient of CO2 is much more complex than that of SF6. Therefore, the citation was limited to Gloor et al. 2007 who report on observations of SF6.

Page 11: TransCom-CH4 experiment (Patra et al. 2011) found dependency of the lag time with emission patterns and rates. Are the differences between the campaigns arising from emission dependency or the interannual viability in transport itself ?

Campaigns were chosen to have a large latitudinal coverage and to include a large portion of data without selections of special meteorological features, but took place at different locations and times. Thus, inter-annual variability in transport certainly contributed to the differences. In addition, the deployed instrumentation has different sampling characteristics, which must not be neglected and is mostly reflected in the larger variability of the higher resolution data. Taking this into account, data from the different campaigns agree well and we are confident that data can be combined and evaluated as a whole.

Line 240 : It would have been good to check using the model experiments in TransCom-CH4, to probe the trends in inter-hemispheric exchange rates.

We agree that such an investigation of the trends in interhemispheric transport would be scientifically valuable, and this has also been suggested by Krol et al. (2018) in their TransCom AoA study. For the current study, deliberately a very simple model was chosen to investigate the potential of upper tropospheric observations as an additional constraint for estimates of global emissions and interhemispheric transport. Analysing the transport trends from the complex models participating in the TransCom-CH4 experiment should certainly be addressed in separate study.

Figure 5 (tropics north; green triangle), and line 9/10: are they consistent ?

The initial wording in lines 9/10 was indeed not fully consistent and has been changed to:

" (…) lag times decrease over the period 2006-2020 in the extra-tropics and the southern tropics."

Finally, regarding the data availability: will it be possible to create a dataset for all the aircraft campaigns with troposphere flagging for further research. I see that as a useful outcome of this paper.

We agree that the dataset is one important outcome of the study. To make it usable in further studies, we followed the reviewer's suggestion and, the dataset was made available publicly with doi 10.5281/zenodo.10018398 and may be downloaded from https://zenodo.org/records/10018398. This information including the URL was added to the data availability statement of the manuscript.

**References:**

Bischof, W., R. Borchers, P. Fabian, and B. C. Krüeger, 1985: Increased concentration and vertical distribution of carbon dioxide in the stratosphere, Nature, 316, 708-710, doi:10.1038/316708a0.

Gloor, M., Dlugokencky, E., Brenninkmeijer, C., Horowitz, L., Hurst, D. F., Dutton, G., Crevoisier, C., Machida, T., and Tans, P. (2007), Three-dimensional SF6 data and tropospheric transport simulations: Signals, modeling accuracy, and implications for inverse modeling, J. Geophys. Res., 112, D15112, doi:10.1029/2006JD007973.

Hall, T. M., and Waugh, D. W. (1998), Influence of nonlocal chemistry on tracer distributions: Inferring the mean age of air from SF6, J. Geophys. Res., 103(D11), 13327–13336, doi:10.1029/98JD00170.

Krol, M., de Bruine, M., Killaars, L., Ouwersloot, H., Pozzer, A., Yin, Y., Chevallier, F., Bousquet, P., Patra, P., Belikov, D., Maksyutov, S., Dhomse, S., Feng, W., and Chipperfield, M. P.: Age of air as a diagnostic for transport timescales in global models, Geosci. Model Dev., 11, 3109–3130, https://doi.org/10.5194/gmd-11-3109-2018, 2018.

Nakazawa, T., K. Miyashita, S. Aoki, and M. Tanaka, 1991: Temporal and spatial variations of upper troposphere and lower stratospheric carbon dioxide, Tellus, Ser. B, 43, 106–117. https://doi.org/10.1034/j.1600-0889.1991.t01-1-00005.x

Patra, P. K., Houweling, S., Krol, M., Bousquet, P., et al., 2011: TransCom model simulations of CH4 and related species: linking transport, surface flux and chemical loss with CH4 variability in the troposphere and lower stratosphere, Atmos. Chem. Phys., 11, 12813–12837. https://doi.org/10.5194/acp-11-12813-2011.

Kovács, T., Feng, W., Totterdill, A., Plane, J. M. C., Dhomse, S., Gómez-Martín, J. C., Stiller, G. P., Haenel, F. J., Smith, C., Forster, P. M., García, R. R., Marsh, D. R., and Chipperfield, M. P.: Determination of the atmospheric lifetime and global warming potential of sulfur hexafluoride using a three-dimensional model, Atmos. Chem. Phys., 17, 883–898, https://doi.org/10.5194/acp-17-883-2017, 2017.

---

## Author Comment (AC2)

**Schuck et al., The interhemispheric gradient of SF6 in the upper troposphere**

**Response to comments by Anonymous Referee #2**

We thank the reviewer for their comments and suggestions to improve our manuscript. All comments are addressed in the following with the reviewer's comments printed in blue, and the responses in black.

This paper uses in situ trace gas measurements, primarily SF6, from the upper troposphere and surface along with a 12-box model to investigate tropospheric transport and SF6 emissions over a recent 15 year period. The combination of US and European based aircraft data sets provides a relatively complete latitudinal and temporal representation of the upper troposphere over this time period. The authors optimize both emissions and transport parameters within the 12-box model to best match the observations resulting in a number of interesting findings. This is a really nice use of a simplified model to diagnose what the observations can tell us about large scale features of atmospheric transport and trace gas emissions. The methodology and discussion of results are clearly described. I recommend publication in ACP with consideration of the minor comments below.

Specific comments:

Section 4 or 5: It would be nice to include a brief comparison of previous model estimates of the interhemispheric transport time such as from Waugh (2013) and Orbe (2016, 2021) to the original estimates from the 12 box model. The Waugh (2013) transport time was less than 2 years to the SH but still somewhat longer than the observational transport time. This comparison would help give the reader an idea of how much CTM transport needs to be adjusted to better match the observations.

We agree with the reviewer's suggestion. Following the comparison of modelled and observed lag times discussed using Figure 6, the following paragraph was added in section 4:

*„Similar results were obtained previously using the more sophisticated NASA Global Modeling Initiative chemical transport model (CTM) (Strahan at al. 2007, 2016). Waugh et al. (2013) compared the CTM output to ground-based, ship-borne, and aircraft measurements from the NOAA observational network and found the model to overestimate lag-times towards southern latitudes. At middle and high latitudes in the southern hemisphere, ground station observations yielded lag-time of 1.3-1.4 years, whereas the CTM results were around 1.75 years for latitudes south of 30° S. Analysing transit time distributions derived from CTM results, Orbe et al. (2016) obtained modelled mean tropospheric age values of 1.5-2 years from the surface up to 200 hPa in the southern extra-tropics. Comparing results of a newer model run to surface observations, Orbe et al. (2021) reported good agreement between surface observations and the model results in the northern hemisphere, but an increasing overestimation by the model towards southern latitudes. In the southern extra-tropics observation-based lag-times of approximately 1.5 years were significantly below the model result of approximately 2 years. The overestimation was largely attributed to the influence of high-$SF_6$ sites on the reference time series used to calculate the model lag times. This supports our choice of using the marine boundary layer zonal average as the reference time series."*

Line 165: Related to the above comment, it might be helpful to briefly state how the initial values of T were obtained for those not familiar with the Rigby 2013 paper. For instance,

*were they based on reanalysis output or a best fit to observed mixing ratios of some trace gases?*

The initial transport parameters $T_{ij}$ were constrained by observations of anthropogenic gases such as chlorofluorocarbons. They are assumed to be applicable for gases with similar emission characteristics, a condition which is fulfilled for SF6 (Cunnold et al. 1983, 1997, 2002).

For the revised manuscript, the related paragraph was extended and now reads:

*„Transport between the individual boxes is realised by parametrising bulk advection and eddy diffusion with the latter term dominating the transport (Cunnold et al. 1983, 1994). This is done with a transport matrix T, with elements $T_{ij}$, that quantify transport between pairs of individual boxes. These transport parameters vary seasonally but have no inter-annual variability. They were derived based on the best fit to chlorofluorocarbon measurements from the AGAGE observational network and can be assumed to yield reasonable results for gases with similar emission characteristics, which holds for gases emitted mainly from anthropogenic sources such as $SF_6$ (Cunnold et al. 1983, 1997, 2002).“*

*Lines 248-9: The difference between the PBL and UT gradient is also much smaller in the model compared to the observations. That seems worth pointing out here.*

We agree and added the following statement to the revised manuscript:

*„In addition, the difference between the surface and the upper troposphere is smaller in the model than observed (cf. Fig. S5).“*

*Line 261: 'were performed'*

Changed as suggested.

**References**

Cunnold, D. M., Weiss, R. F., Prinn, R. G., Hartley, D., Simmonds, P. G., Fraser, P. J., Miller, B., Alyea, F. N., and Porter, L. (1997), GAGE/AGAGE measurements indicating reductions in global emissions of $CCl_3F$ and $CCl_2F_2$ in 1992–1994, *J. Geophys. Res.*, 102(D1), 1259–1269, doi:10.1029/96JD02973.

Cunnold, D. M., et al., In situ measurements of atmospheric methane at GAGE/AGAGE sites during 1985–2000 and resulting source inferences, *J. Geophys. Res.*, 107(D14), doi:10.1029/2001JD001226, 2002.

Strahan, S. E., Duncan, B. N., and Hoor, P.: Observationally derived transport diagnostics for the lowermost stratosphere and their application to the GMI chemistry and transport model, Atmos. Chem. Phys., 7, 2435–2445, https://doi.org/10.5194/acp-7-2435-2007, 2007.

Strahan, S. E., Douglass, A. R., and Steenrod, S. D. (2016), Chemical and dynamical impacts of stratospheric sudden warmings on Arctic ozone variability, *J. Geophys. Res. Atmos.*, 121, 11,836–11,851, doi:10.1002/2016JD025128.

---

## Author Comment (AC3)

**Response to comments by Anonymous Referee #3**

We thank the anonymous reviewer #3 for their comments and for their suggestions to improve the manuscript. All comments are addressed in the following with the reviewer's comments printed in blue, and the responses in black.

General comments

This study combined upper tropospheric (UT) SF6 datasets from different measurement projects and attempted to interpret them in terms of emissions and transport. Such efforts to utilize datasets from multiple projects are of particular importance from long-term point of view. I think that a follow-up study using a 3D transport model is needed, but this manuscript has made a good job to motivate such future studies. Before agreeing to publication of this study, I would like to encourage the authors to consider my comments below to enrich scientific discussion and provide better guidance to following studies.

Orbe et al. (2021) presented detailed analyses of the SF6 age (or lag time as called in this study) including the data from the ATom campaign. This study repeats analysis of the ATom data, and extends to CARIBIC, GhOST and HIPPO, therefore number of the data in the UT is extensively increased. However, spatial distributions of the SF6 age presented in this study is Figure 4 only. With the data newly analyzed here, the authors could provide more presentations about spatial variations of the SF6 age like Figures 4 and 5 in Orbe et al. (2021). In particular, using the CARIBIC data, the authors might discuss longitudinal variation of the SF6 age over NH, which could reflect distribution of the underlying sources as well as areas of effective vertical transport. Also, the authors might confirm (or discuss difference of) the vertical gradient of the SF6 age over different latitudinal regions in comparison to Orbe et al. (2021). I think that such more in-depth analysis would highlight value of this study and further motivate follow-up studies using a 3D transport model.

In contrast to the study by Orbe et al. (2021) our study excludes stratospheric data and measurements at low altitudes. Time lags derived from a reference time series differ considerably for tropospheric and stratospheric data and to account for the different transport pathways, different reference time series would have to be chosen for tropospheric and stratospheric observations. Low altitudes were excluded because they are not covered by CARIBIC data. Therefore, a Figure similar to Fig. 5 of Orbe et al (2021) showing altitude profiles is not included.

We appreciate the suggestion of showing a spatial distribution of the derived lag times and have included a figure showing the spatial distribution of the measurement locations, colour-coded by lag-time in the revised version of the supplementary document. A corresponding figure reference has been added to the main text in section 3:

*"The spatial distribution of lag-times is shown in the map in the supplementary Fig S1."*

One important question (but I cannot find the answer clear) in this study is whether the SF6 age in the UT of SH has a secular decreasing trend, in contrast to Orbe et al. (2021) who

showed negative trends of the SF6 age south of 30°N at the surface. In abstract and conclusion, the authors argue that it exists, but they describe it a bit more carefully in section 3. As the authors writes, data are relatively limited in SH, so the trend line shown in Figure 5 could be unrepresentative (not free from sampling bias). It would be interesting to see the authors' argument more clearly about whether they regard the trend significant/well represented or still difficult due to limited availability of data. Regarding this, I wonder about possible use of the modeling results. The authors optimized the model (though it was tuned by more number of NH data). Figure 8 indicates almost constant value of data minus model with time in the UT exT-S. Does it mean that the model consistently shows a negative trend? What if the authors add the optimized model trend line to Figure 5? Thinking that the additional constraints from the newly analyzed data seem to be value of this study, I would like to see extended presentation of the outcome (e.g., modified Figure 5 or new figure after Figure 8) to answer the question.

As the southern hemisphere extra-tropics are clearly under-represented in the dataset we prefer to stay with the careful wording about the trends. As pointed out by another reviewer, the abstract wording and Figure 5 were not consistent regarding the southern hemisphere. The abstract wording about the trends has been changed to:

*"At the most southern latitudes, a lag time of over 1 year with respect to the northern mid-latitude surface is derived, and lag times decrease over the period 2006-2020 in the extra-tropics and the southern tropics."*

In the main text, we follow the reviewer's suggestion to also discuss the trends found in the modelled lag times. Restricting the trend analysis to the years in which observational data are available in the respective latitude band, all trends are negative in the model output, but are statistically insignificant (at the 2-sigma level) except for the northern extra-tropics. There we find a negative trend of -0.07 years per decade (±0.02) which is much smaller than what is derived from the observations (-0.14 ± 0.05  yrs/dec). The following section was added in section 4:

*"Fitting a trendline in each latitude band over the time period covered by observations as visible in Fig. 5, modelled trends are negative in all four upper tropospheric boxes, but are not statistically significant (2 σ). Only in the northern extra-tropics (Box 4), a statistically significant trend of -0.07±0.02 years/decade is obtained, smaller than the value of -0.14±0.05 years/decade in the observations."*

Figure 5 was left unchanged, because adding four more trend lines would make it more difficult to extract information from the figure. Given that of these only one line has a slope significantly different from zero, having model results in the figure would add only minor information.

Minor comments

Introduction: Interhemispheric transport with surface emissions mainly in the NH mid latitudes shapes latitudinal gradient of trace gas of interest. Citing previous studies, the authors highlight more effective interhemispheric transport in the UT than near the surface. Many

trace gases with major emissions in the NH (e.g., CO2, CH4, CO, SF6 etc.) shows smaller latitudinal gradient in the UT than near the surface, but this cannot be explained by different efficiency of interhemispheric transport in different vertical layers only. I think that discussion on upward air inflow from surface to the UT over the tropics (the major pathway of surface air into the UT), is missing despite that it also mitigates latitudinal gradient in the UT. Japanese groups have made significant contributions to interpretation of latitudinal gradients of CO2 and related gases in the UT and near the surface (Nakazawa et al. 1991; Miyazaki et al. 2008; Sawa et al. 2012; Bisht et al. 2021). The authors also mention to the Asian summer monsoon. Important in the context of this study is probably that Asian summer monsoon effectively uplift South Asian surface air ("polluted" air) to the UTLS, the gateway to interhemispheric transport as the authors cite previous studies like Yan et al. (2021) and Belikov et al. (2022). Such an express pathway from NH low latitude surface to the UT of SH also weakens the latitudinal gradient in the UT. I think that the introduction could be reformulated so as to inform readers about supply of surface air into the UT by various pathways also play an important role in determining the latitudinal gradient in the UT.

The statement on vertical transport in the introduction has been extended to better account for other transport pathways by adding the following text:

*"The parametrization of convection and thus vertical transport from the surface to higher altitudes was identified as an important factor for differences of interhemispheric transport time between models (Orbe et al. 2018, Krol et al. 2018).*

*While the interhemispheric gradient of SF6 is mainly driven by the interhemispheric asymmetry of surface emissions, transport pathways from the surface to the upper troposphere also influence the latitudinal variation of mixing ratio at altitude (Miyazaki et al. 2009). In particular tropical convection can rapidly bring air masses with elevated mixing ratios of SF6 and other tracers as for example from the Asian monsoon region or over tropical Africa to the upper troposphere (e. g. Randel and Park, 2006; Schuck et al., 2010; Vogel et al., 2016; Thorenz et al., 2017). Convection over remote marine tropical regions in contrast results in an inflow of air with low mixing ratios of anthropogenic tracers. This also implies that the interhemispheric gradient of SF6 could vary with longitude as observed for example for CH4 analysing measurements in the upper troposphere from IAGOS-CARIBIC (In-Service Aircraft for a Global Observing System - Civil Aircraft for the Regular Investigation of the Atmosphere Based on an Instrument Container) and CONTRAIL (Comprehensive Observation Network for TRace gases by AIrLiner) flights into the southern hemisphere (Schuck et al., 2012)."*

(Initial wording was *"The parametrization of convection and thus vertical transport from the surface to higher altitudes was identified as an important factor for differences of interhemispheric transport time between models (Orbe et al. 2018, Krol et al. 2018)."*)

P5 L114: I think that the original measurements were not made on the WMO X2014 scale (officially the scale is named without "NOAA"). It would be better to describe about the original measurement scale and the conversion applied retrospectively.

When revising the manuscript we changed the scale name to WMO 2014 (skipping "NOAA") as suggested. In additional the following statement was added:

*"Data measured prior to the publication of this scale were converted from the WMO 2006 scale to the WMO 2014 scale, both maintained by NOAA ([https://gml.noaa.gov/ccl/sf6_scale.html](https://gml.noaa.gov/ccl/sf6_scale.html)). This was typically a correction of 0.01 ppt or less"*

Table 1: The precision of SF6/N2O could be given in ppt/ppb, to be consistent with the following discussions. Use of ppb/ppbV is confusing. I think that the authors distinguish them on purpose, but no explanation is given. For instance, TRIHOP CO is in ppbV while UMAQS CO is in ppb despite the same measurement principle. This should be sorted out or consistent explanation should be given.

We agree that usage of ppb/ppbV was inconsistent and now use ppb. The choice of using absolute (ppt/ppb) vs relative (%) precisions depends on how these numbers are derived. For CARIBIC data, it is therefore more appropriate to report relative precisions, also for comparison with the numbers published earlier. For comparison within Table 1, absolute numbers in ppt/ppb were added.

P7 L135: "Because of their higher time resolution" high resolution does not mean observation of large variability. High-resolution measurements could capture phenomena that could not be resolved by low-frequency sampling, but underlying nature is that large variability is happening at place of interest. For instance, SF6 measurements at a very remote site (e.g., Antarctica) show small variability even if measured at high frequency.

We agree with the reviewer that the underlying variability of observations depends on the sampling location. As pointed out by the reviewer, high-resolution measurements may capture features which are unresolvable for methods with lower time resolution such as air sample collection, which in most cases averages over longer sampling times than online measurements. For observations in the same location this will lead to a higher variability in high-resolution data, therefore we think the above statement is correct and largely explains the higher variability of the research aircraft mission in comparison to the IAGOS-CARIBIC data despite their different spatial coverage.

Figure 2: The reference SF6 time series could be added to visualize the concept of the SF6 lag time.

The reference time series of SF6 has been added to Figure 2 in the revised manuscript and the figure caption was extended correspondingly.

Section 2.3: Orbe et al. (2021) pointed out that different choice of the reference time series could result in different trend of the SF6 lag time. The authors could discuss how this affects and why the simple marine boundary layer choice is made in this study.

This is now discussed in section 4, where the following paragraph was added:

*„Similar results were obtained previously using the more sophisticated NASA Global Modeling Initiative chemical transport model (CTM) (Strahan at al. 2007, 2016). Waugh et al. (2013) compared the CTM output to ground-based, ship-borne, and aircraft measurements from the NOAA observational network and found the model to overestimate lag-times towards south-*

*ern latitudes. At middle and high latitudes in the southern hemisphere, ground station observations yielded lag-times of 1.3-1.4 years, whereas the CTM results were around 1.75 years for latitudes south of 30° S. Analysing transit time distribution derived from CTM results, Orbe et al. (2016) obtained modelled mean tropospheric age values of 1.5-2 years from the surface up to 200 hPa in the southern extra-tropics. Comparing results of a newer model run to surface observations, Orbe et al. (2021) reported good agreement between surface observations and the model results in the northern hemisphere, but a decreasing overestimation by the model towards southern latitudes. In the southern extra-tropics observation-based lag-times of approximately 1.5 years were significantly below the model result of approximately 2 years. The overestimation was largely attributed to the influence of high-$SF_6$ sites on the modelled reference time series used to calculate the model lag times. This supports our choice of using the marine boundary layer zonal average as the reference time series."*

Figure 3: It would be interesting to see latitudinal distributions from surface sites along with those in the UT.

When preparing new figures during manuscript revision, the surface mixing ratios for the latitudes outside ±30° of each hemisphere were added for those months for which CARIBIC data is shown in Fig. 3.

Figure 4: As in my earlier comment, I hope to see more in-depth analyses of the SF6 lag time, not only for latitude but for longitude and altitude.

The longitudinal variation is briefly touched on in the discussion of Fig 3. We've extended this by adding the following statement:

*„Thus, the latitudinal gradient also has a longitudinal variation depending on emission and transport patterns."*

Discussion of Fig. 4 was changed and extended to:

*"All four data sets agree with each other within their respective variability. In particular, this is the case for data from the HIPPO flight which cover a longitude range that is under-represented in the CARIBIC data and not covered by the included HALO missions. From this it can be concluded, that the longitudinal variability is smaller than the interhemispheric gradient. Thus, all observations are combined applying the above filter procedure to all observations simultaneously and zonal averages are discussed in the following."*

The dataset we use is restricted to measurements at pressure below 400hPa, excluding stratospheric data, covering a very small altitude range. We agree that a discussion of the vertical distribution of SF6 mixing ratios would be interesting to look at, and we plan to do so in a future study. While the current study looks at interhemispheric transport in the troposphere, studying the vertical gradient of SF6 should also include stratospheric data and discuss cross-tropopause transport. This is not possible without using a sophisticated model and thus beyond the scope of the current study.

P11 L195: As I commented to introduction, the smaller latitudinal gradient in the UT than

near the surface is not fully attributable to active interhemispheric transport in the UT but is also contributed by intrusion of surface air into the UT.

For the revised manuscript, the introduction was extended to account for this aspect. The here mentioned statements about consistency with the results obtained by Belikov et al. (2022) was left unchanged.

Figure 5: As in my earlier comment, this could be compared to results of the optimized model output. The trend line for ex-tropics south is hard to see in its color. Labels could be larger in size.

As the trends in the model output reflect the prescribed trend from the EDGAR emission flux time series, we deliberately do not discuss trends of the model output but use the detrended metric of the lag time for all comparisons.

When preparing new figures during manuscript revision, the label size was increased and a darker colour was chosen.

Section 4: The authors mention to possible southward shift of SF6 emissions in the NH as discussed in previous studies (e.g. Orbe et al. 2021). I think that the authors could have made an experiment in which latitudinal emission patterns change with time in their box modeling. Yang et al. (2019) pointed out that vertical transport in the extratropics of NH (T04 in this study) is also important for north-south gradient of the SF6 age. I wonder whether the authors have made sensitivity tests for this parameter. The authors argue that "horizontal transport seems to be more important than vertical transport" (P18 L333), at which I wondered about inconsistency with Yang et al. (2019).

In their sensitivity studies Yang et al. (2019) compared the sensitivity of tropospheric age ($a_{SF6}$ in their nomenclature) for which we prefer the term "lag time" and of the interhemispheric exchange time $\tau_{ex}$. They concluded that the age a is more sensitive than the exchange time $\tau$ to perturbations of the vertical exchange between lower troposphere and upper troposphere over the northern extratropical source region (experiment S2) in their study. They obtain a similar result perturbing the transport from the northern extratropical source to the northern tropics.

Certainly the main difference between the sensitivity experiment performed by Yang et al (2019) and our current study is the multi parameter approach in which we vary all horizontal transport parameters simultaneously. We agree that vertical transport does also play a role. However, from the time lag differences between the model and the observations for neighbouring boxes presented in Fig. S4 and S5 we conclude that the vertical transport parameters have a smaller impact on modelled lag times (corresponding to tropospheric a in the Yang et al (2019) study) than the horizontal transport parameters and therefore restricted the sensitivity study to the latter.

Adding a time variation of the emission fluxes (beyond the time variation inherent to the EDGAR emission fluxes time series) would introduce an even higher degree of complexity to the sensitivity runs. A time change of the emissions would of course be reflected in modelled

mixing ratios, but as laid out in section 4, these are strongly influenced by the initialization values of the model run.

Reference

Nakazawa et al. (1991) https://doi.org/10.1034/j.1600-0889.1991.t01-1-00005.x

Miyazaki et al. (2008) https://doi.org/10.1029/2007JD009557

Sawa et al. (2012) https://doi.org/10.1029/2011JD016933

Bisht et al. (2021) https://doi.org/10.1029/2020JD033541

Miyazaki, K., Machida, T., Patra, P. K., Iwasaki, T., Sawa, Y., Matsueda, H., and Nakazawa, T. (2009), Formation mechanisms of latitudinal CO2 gradients in the upper troposphere over the subtropics and tropics, J. Geophys. Res., 114, D03306, doi:10.1029/2008JD010545.

Randel, W. J., and M. Park (2006), Deep convective influence on the Asian summer monsoon anticyclone and associated tracer variability observed with Atmospheric Infrared Sounder (AIRS), J. Geophys. Res., 111, D12314, doi:10.1029/2005JD006490.

Schuck, T. J., Brenninkmeijer, C. A. M., Baker, A. K., Slemr, F., von Velthoven, P. F. J., and Zahn, A.: Greenhouse gas relationships in the Indian summer monsoon plume measured by the CARIBIC passenger aircraft, Atmos. Chem. Phys., 10, 3965–3984, https://doi.org/10.5194/acp-10-3965-2010, 2010.

Vogel, B., Günther, G., Müller, R., Grooß, J.-U., Afchine, A., Bozem, H., Hoor, P., Krämer, M., Müller, S., Riese, M., Rolf, C., Spelten, N., Stiller, G. P., Ungermann, J., and Zahn, A.: Long-range transport pathways of tropospheric source gases originating in Asia into the northern lower stratosphere during the Asian monsoon season 2012, Atmos. Chem. Phys., 16, 15301–15325, https://doi.org/10.5194/acp-16-15301-2016, 2016.

---

## Author Response (AR2)

**Schuck et al., The interhemispheric gradient of SF₆ in the upper troposphere**

**Response to comments by Editor Farahnaz Khosrawi**

We thank the editor for her comments on the revised manuscript. The suggestions to further improve the wording are greatly appreciated, and the manuscript was modified as follows (editors' comments in blue, resposnes in black).

P3, L45: "at altitudes" -> which altitudes? Is here something missing?

We agree that the initial wording "at altitude" is not very precise and have changed this to "at aircraft cruise altitudes of 8-13 km".

P7, L149: which average -> which are averaged

As "which are averaged" might be mistaken as a mathematical averaging, the wording was changed to "which represent an average mixing ratio over the sampling time of 30—240 s"

P10, L197: I am not sure if just writing "into" is correct here. Wouldn't it be better to write from "the northern hemisphere to the stratosphere" (thus to state from where these flew into the southern hemisphere).

"flights into the southern hemisphere" was changed to "flights with sections in the southern hemisphere". The same wording was used on P4, L53 and the caption of Fig. 3 and was changed there correspondingly.

P10, L219: "data from the HIPPO flight" -> also here I would suggest to more clearly write "data from the flight from the HIPPO campaign".

The wording "data from the HIPPO flight" (which should have been "flights") was changed to "data from the HIPPO missions".

P12, L230: Fig S1 -> Fig. S1

Corrected.

P13, L252: I would suggest to make to sentences. Thus fullstop after "here" and new sentence starting with "One" or to use a simicolon.

Changes as suggested.

P14, L273 and 278: Reference to Fig. S5 before Fig. S2. Supplementary figures should also be referenced in a consecutive order.

Corrected to references in consecutive order.

P15, L295: "the" before "reference time series" obsolete?

Is seems appropriate to use "the" here to stress that there is one particular reference time series used.

P15, L313: Check sentence, something is wrong here. I would skip "As" and start the sentence "The results shown in Fig 6. indicate..." or "The time lag shown in Fig. 6 indicate...".

The wording has been changed to: "The results shown in Fig. 6 indicate that in particular emissions at southern latitudes might be too low. To test this, up to 28 % of emissions were taken out of the northern hemisphere extra-tropics (Box 0) and shifted southward into the tropics (Boxes 1 and 2)."

P17, L340: Also this sentence need to be checked. Something is wrong here, too. How can evaluating the lag time improve the results?

The MAD as a quantifier for the model-observation deviation can be evaluated based on time lags or based on mixing ratios. Here, we compare the improvement obtained for both. To make this more clear the sentence was reworded to:

"The value of $d(MAD)_{mxr}$ improves to 0.18 ppt for this model setup. Evaluating the differences between modelled and observed lag times, $d(MAD)_{lag}$ improves to 0.47 years compared to 1.95 years with the default transport scheme and unscaled emissions."

P18, L356 and L361: AGAGE box model already mentioned before introduced. Provide the details on model and the reference already in line 356.

L 356 has been changed and does not refer to the model anymore.

P19, L370: Check also this sentence and consider to split in several sentences or to better connect the text parts.

The lengthy passage was cut into several sentences and now reads: "To study the influence of the model transport scheme, a series of sensitivity runs was performed. Thereby better agreement was found with a 3.25\,\% global emission increase relative to EDGAR 7 bottom-up emissions in combination with a southward shift of emissions and a modified transport scheme. The latter combined a weaker tropical transport barrier and thus faster transport into the southern hemisphere."

P19, L377: slower transport of what? Please be more precise.

Wording was extended to "slower southward transport".

Reference list: Please check, that the style is consequently done according to the Copernicus style. In some cases titles start all words with capital letters, in some cases the normal upper/lower case writing is used.

We've used to original upper/lower case writing used by the original journal and will change this during the final typesetting according to the recommendations from the language editing team.

In some cases subscripts for the chemical species as e.g. SF6 are missing etc.

All missing subscripts have been corrected.

Further language simplifications:

P 3, L32: Split sentence to "Model results indicate that interhemispheric transport is asymmetric, with transport from the northern into the southern hemisphere being faster than vice versa. This influences the north-south gradient and interhemispheric transport times (Krol et al., 2018).

P 3, L45: mixing ratio -> mixing ratios

P3, L 47: Reworded to: "In particular tropical convection can rapidly bring air masses with elevated mixing ratios of $SF_6$ and other tracers to the upper troposphere,  for example from the Asian monsoon region or over tropical Africa (e. g. Randel and Park, 2006; Schuck et al., 2010; Vogel et al., 2016; Thorenz et al., 2016). Convection over remote marine tropical regions, in contrast, results in an inflow of air with low mixing ratios of anthropogenic tracers. This also implies that the interhemispheric gradient of $SF_6$ could vary with longitude. This was observed for example for $CH_4$ in the upper troposphere during IAGOS-CARIBIC (In-Service Aircraft for a Global Observing System - Civil Aircraft for the Regular Investigation of the Atmosphere Based on an Instrument Container) and CONTRAIL (Comprehensive Observation Network for TRace gases by AIrLiner) flights with sections in the southern hemisphere (Schuck et al., 2012)."

P5, L85: Dropped repetition of "series".

P7,144: comma added

P9, L190: Sentence split into: "This value was derived as the average offset of the lowermost stratosphere with regard to the upper troposphere from the cross-tropopause gradient of the CARIBIC data set. Data from the northern hemisphere mid-latitudes with a potential temperature difference of 5 K above the thermal tropopause were used."

P 10, L210: Removed  "which corresponds to all upper tropospheric observations"

P14, L292: comma added

P15, L301 and 302 and 303: comma added